# L3Ms — Lagrange Large Language Models

**Guneet S. Dhillon [1]\*, Xingjian Shi [2], Yee Whye Teh [1], Alex Smola [2]**
[1] University of Oxford, [2] Boson AI
{guneet.dhillon,y.w.teh}@stats.ox.ac.uk, {xingjian,smola}@boson.ai

## Abstract

Supervised fine-tuning (SFT) and alignment of large language models (LLMs) are key steps in providing a good user experience. However, the concept of an appropriate alignment is inherently application-dependent, and current methods often rely on heuristic choices to drive optimization. In this work, we formulate SFT and alignment as a constrained optimization problem: the LLM is fine-tuned on a task while being required to meet application-specific requirements, without resorting to heuristics. To solve this, we propose Lagrange Large Language Models (L3Ms), which employ logarithmic barriers to enforce the constraints. This approach allows for the customization of L3Ms across diverse applications while avoiding heuristic-driven processes. We experimentally demonstrate the versatility and efficacy of L3Ms in achieving tailored alignments for various applications.

## 1 Introduction

Large language models (LLMs) are used for a wide range of tasks: as chatbots (Brown et al., 2020; OpenAI, 2024), for code generation (Ahmad et al., 2021; Wang et al., 2021; Rozière et al., 2024), for medical assistance (Yang et al., 2022; Moor et al., 2023), and so on. The key ingredients for their impressive downstream performance are supervised fine-tuning (SFT) and alignment; the former fine-tunes the LLM to a task of interest, while the latter instills it with preferential properties. Arguably, the *right* combination of preferential properties is highly application-dependent. For example, a scholar would want a chatbot be honest and factual for assistance with their work. In contrast, a fiction writer might prefer the opposite behavior to help create fantastical imaginary worlds. There is also plenty of (anecdotal) evidence in support: some LLMs refuse to provide information on how to "kill" a process in Unix, recommending the use of less violent strategies for dealing with wayward computer programs instead.[1] Therefore, we need frameworks for LLM customization.

Consequently, Bai et al. (2022); Rame et al. (2023); Wu et al. (2023); Ji et al. (2023); Zhou et al. (2024) fine-tune LLMs on varying combinations of such preferential properties. In practice, one tends to resort to trial and error to find the right combination of preferences for their particular application. In doing so, one verifies if certain *minimum baselines* are satisfied, such as ensuring the factual correctness of statements or confirming that response lengths are capped at 100 words. Since there isn't a way to enforce such requirements directly, current methods resort to heuristics. Additionally, existing pipelines carry out SFT and alignment sequentially and must ensure that the LLM does not forget relevant task information learned during the SFT stage. This is achieved by penalizing the LLM for drastic deviations, with the strength of the penalty determined heuristically.

In this work, we formulate SFT and alignment in LLMs as a constrained optimization problem. In particular, we fine-tune an LLM to minimize the task objective (the objective function) while simultaneously satisfying application-specific minimum requirements (the constraints). This merges the SFT and alignment stages and mitigates the reliance on heuristics altogether. Furthermore, we propose Lagrange Large Language Models, a.k.a. L3Ms, to solve such constrained optimization problems. Specifically, we do so by employing logarithmic barriers and gradually enforcing the constraints during the fine-tuning procedure. Lastly, we empirically demonstrate how one can pick and choose constraints and tailor L3Ms to a range of applications without resorting to heuristics.

---

\*Work done while at Boson AI

[1] https://www.reddit.com/r/LocalLLaMA/comments/180p17f/new_claude_21_refuses_to_kill_a_python_process/

| Prompt | What are the four main types of macromolecules found in living organisms? |
|---|---|
| **Response** 
 **(short response L3M)** | 1. Nucleic acids (DNA and RNA) 
 2. Proteins 
 3. Carbohydrates 
 4. Lipids |
| **Response** 
 **(long response L3M)** | 1. Nucleic acids, including DNA and RNA, which carry genetic information 
 2. Proteins, which are responsible for the structure, function, and regulation of living organisms 
 3. Lipids, which provide energy and form the membranes that surround cells 
 4. Carbohydrates, which are used as an energy source and as structural components of cells. |
| **Prompt** | Who discovered the law of universal gravitation? |
| **Response** 
 **(short response L3M)** | Sir Isaac Newton is credited with discovering the law of universal gravitation in the late 17th century. |
| **Response** 
 **(long response L3M)** | Sir Isaac Newton is widely credited with the discovery of the law of universal gravitation, which states that every particle in the universe attracts every other particle with a force proportional to their masses and inversely proportional to the square of the distance between them. |

Table 1: **Example responses from length constrained L3Ms.** We provide example responses from L3Ms with varying length constraints. We include the prompt along with the generated responses from two L3Ms; one constrained to have short responses and the other constrained to long ones.

For example, Table 1 provides the generated responses from two such L3Ms; both are fine-tuned for instruction-following, but one is constrained to be concise, while the other is to be verbose.

In summary, our contributions are as follows:

1. We formulate SFT and alignment in LLMs as a constrained optimization problem: an LLM is fine-tuned on a task while simultaneously satisfying custom requirements (cf. Section 4).
2. We propose L3Ms, a family of LLMs fine-tuned using the above framework (cf. Section 5).
3. We experimentally demonstrate how L3Ms can be customized to different applications and their specific requirements while avoiding heuristic-driven processes (cf. Section 6).

## 2 OPTIMIZATION FOR LLMS

Training an LLM proceeds in multiple stages (Ouyang et al., 2022), which we discuss below.

### 2.1 PRE-TRAINING

The pre-training stage instills the LLM with a generic knowledge of language. It entails regenerating text/token sequences by minimizing their perplexity, i.e., the negative log-likelihood of the sequence normalized by its length. More formally, the perplexity on a sequence $x$ is defined as:

$$l_\theta\left(x\right) \; = \; -\frac{\log \pi_\theta\left(x\right)}{|x|} \; = \; -\frac{1}{|x|}\sum_{i=1}^{|x|}\log \pi_\theta\left(x_i|x_{<i}\right),$$

where $x_i$ and $x_{<i}$ denote the $i$-th token in the sequence and its prefix, respectively. Additionally, the function $\pi_\theta(\cdot)$ denotes the predicted probability distribution of the LLM over token sequences, where the LLM is parameterized with weights $\theta$. Then, the pre-training objective is given as:

$$\min_\theta \; \mathbb{E}_{x\sim q(\cdot)}\left[l_\theta\left(x\right)\right],$$

and the expectation is replaced by an empirical average over a large corpus with trillions of tokens.

## 2.2 Supervised fine-tuning (SFT)

Next, one fine-tunes the LLM to a task of interest, such as instruction-following, summarization, or translation. The data are (prompt, response) pairs $(x, y)$, and the LLM is fine-tuned to regenerate the responses. Thus, one minimizes the perplexity (or a related loss) on the response given the prompt:

$$\min_{\theta} \ \mathbb{E}_{(x,y) \sim p(\cdot)} \left[ l_{\theta} \left( y | x \right) \right], \tag{1}$$

for a distribution $p(\cdot)$ over (prompt, response) pairs that reflects the task-related data.

## 2.3 Alignment

This stage aligns the LLM to generate responses with preferential properties. A common setup is to learn preference reward functions that represent properties such as helpfulness and harmlessness (Bai et al., 2022), followed by reinforcement learning to adapt the LLM to maximize the said rewards. This is called reinforcement learning from human feedback (RLHF; Knox & Stone, 2008; Griffith et al., 2013; Christiano et al., 2017). Note that the preference reward functions need not always be learned; they could also be engineered or rule-based, such as the length of the response.

Given a single preference reward function $r(\cdot)$, the alignment objective is given as:

$$\max_{\theta} \ \mathbb{E}_{\substack{(x, \cdot) \sim p(\cdot) \\ y \sim \pi_{\theta}(\cdot | x)}} \left[ r \left( y | x \right) \right]. \tag{2}$$

This maximizes the rewards for the LLM's responses to prompts sampled from the task distribution. To prevent over-optimization of the reward and avoid drastic deviation away from the SFT model, it is common practice to add a regularization penalty, such as the KL divergence (Gao et al., 2023).

We are interested in $k \geq 1$ different preferential properties. In such a scenario, one could learn individual preference reward functions $r_i(\cdot)$'s and optimize the LLM to maximize their combination. In particular, Li et al. (2021); Rame et al. (2023); Wu et al. (2023); Ji et al. (2023) use a linear combination of the rewards, substituting the single reward in Eq. (2) with $\sum_{i=1}^{k} \alpha_i r_i(y|x)$, for some choice of $\alpha_i \geq 0$. Alternatively, one could learn a single reward function by choosing the data proportions of the different properties used to train it. For example, Bai et al. (2022) use a 3:1 proportion of helpfulness to harmlessness data to train a single preference reward function. Note that the proportions can also be represented as weights $\alpha_i \geq 0$ with $\sum_{i=1}^{k} \alpha_i = 1$. Therefore, the choice of $\alpha_i$'s, combined with the strength of the regularization penalty, together steer the alignment. They are chosen based on the good judgment of the practitioner in a somewhat heuristic manner.

## 2.4 Shortcomings

Although the above pipeline is commonly used to train LLMs for deployment, it has shortcomings.

Firstly, how does one choose the weights $\alpha_i$'s? Or equivalently, *what is the right combination of preference properties?* Choosing between properties such as truthfulness, verbosity, and humor depends on the application at hand. However, even when the application is known, the weights are chosen through trial and error: trying different combinations and inspecting the achieved rewards. For instance, if one wants the responses to be under 100 words for a summarization task, one might repeatedly examine the length of the responses and adjust the weights $\alpha_i$'s to achieve that. This inherently involves verification against a set of minimum baselines being satisfied. *Can we avoid heuristics and enforce such minimum requirements for the preference rewards in a principled way?*

Secondly, recall that the original task objective is given in Eq. (1). However, optimizing a special purpose objective such as Eq. (2) can lead to a decrease in performance on the original task. Whilst penalizing the deviation away from the SFT model mitigates this to some extent, the strength of the penalty is again determined heuristically. *Can we ensure the performance on the original task?*

# 3 Related work

## 3.1 One size does not fit all

Early work on LLM alignment assumed homogeneity of preferences (Bakker et al., 2022). However, the reality is quite different: human preferences vary widely and are highly application-specific

(Rame et al., 2023; Casper et al., 2023). Consequently, Bai et al. (2022); Rame et al. (2023); Wu et al. (2023); Ji et al. (2023); Zhou et al. (2024) linearly combine several fine-grained preference rewards, where each combination reflects a unique customization. However, the *right* combination of preferential properties for a particular application is determined heuristically by trial and error. In contrast, we formulate alignment as a constrained optimization problem. In doing so, custom requirements are naturally imposed as constraints. This alleviates the need for heuristic choices.

## 3.2 Constrained alignment for LLMs

Independent of our work, Moskovitz et al. (2024); Dai et al. (2024) also introduced constrained optimization for LLM alignment but for varying reasons. Motivated to avoid over-optimization of preference rewards, Moskovitz et al. (2024) find "proxy points", values of the reward functions beyond which the performance of the LLM is negatively impacted. They constrain the average rewards to be in the vicinity of these proxy points. Dai et al. (2024) trade-off between helpfulness and harmlessness from a safety perspective. They maximize the LLM's helpfulness while constraining its average harmlessness reward. Our motivation, on the other hand, is to tailor to custom preferences through sets of different constraints while simultaneously learning to perform the task of interest.

Our work is different in two important ways. Firstly, both Moskovitz et al. (2024); Dai et al. (2024) consider the alignment process in isolation, because of which their objective is either to maximize one of the preference rewards or to minimize the deviation away from the SFT model. Instead, we merge the SFT and alignment stages and minimize the task objective directly. This ensures that the LLM learns task-solving capabilities without any deviation penalty. Furthermore, by avoiding the computation of the deviations from the SFT model, we save both memory and time during the fine-tuning process: we no longer need to load the SFT model and evaluate it on generated responses.

Secondly, both Moskovitz et al. (2024); Dai et al. (2024) obtain a solution to the constrained problem by computing a saddle point of the Lagrangian. This is achieved by formulating a minimax game where the LLM minimizes the Lagrangian, and the Lagrange multipliers adapt to maximize it. However, the non-convexity of the objective makes this nontrivial. Instead, we choose the logarithmic barrier approach as the associated Lagrange multipliers satisfy the KKT complementary slackness condition by design (cf. Section 5.2) rather than letting the learning procedure do so (which in our experience is extremely sensitive to the choice of the learning rate). We empirically observe that while both methods satisfy the constraints, our approach using log barriers minimizes the task objective better (cf. Section 6.3). As a side effect, we avoid introducing new learning parameters.

Furthermore, we provide experimental validation of our approach using more preferences than Moskovitz et al. (2024); Dai et al. (2024) and $4.5\times$ larger LLMs than Moskovitz et al. (2024).

## 4 Constrained optimization for LLMs

Our goal is to reduce the task objective for the LLM to solve the task of interest while also enabling custom alignment by having the LLM meet the application-specific minimum requirements on different preference rewards. To do so, we propose the constrained optimization problem:

$$\min_{\theta} \ \mathbb{E}_{(x,y)\sim p(\cdot)} \left[ l_\theta \left( y|x \right) \right] \tag{3}$$

$$\text{subject to } \mathbb{E}_{\substack{(x,\cdot)\sim p(\cdot) \\ y\sim\pi_\theta(\cdot|x)}} \left[ r_i \left( y|x \right) \right] \ \geq \ b_i \text{ for all } i \in \{1, 2, \dots, k\}. \tag{4}$$

Here, the objective is the same as that of SFT in Eq. (1), and the constraints are enforced to satisfy the custom requirements; this merges the SFT and alignment stages. The $b_i$'s signify the minimum baselines for each preference reward function $r_i(\cdot)$, and the constraints are enforced on average.

Compared to the previous approach, we no longer rely on heuristics to find a set of weights $\alpha_i$'s to satisfy the minimum requirements; we can do so directly through the constraints. Furthermore, with the same objective as SFT, we can directly maintain task performance without any deviation penalty. Additionally, note that whenever a constraint is satisfied, its influence vanishes. For example, if the LLM is naturally harmless and $r_{\text{harmless}}(y|x) > b_{\text{harmless}}$, then the constraint is not active, and the LLM will not be penalized. In contrast, the previous approach would further penalize the LLM.

**Notation**   For ease of notation, we rewrite the constrained problem in Eqs. (3) and (4) as:

$$\min_{\theta} \ L\left(\theta\right) \ \text{subject to } C_i\left(\theta\right) \ \leq \ 0 \text{ for all } i \in \{1, 2, \ldots, k\}, \tag{5}$$

with the objective $L(\theta) = \mathbb{E}_{(x,y)\sim p(\cdot)}[l_\theta(y|x)]$ and constraints $C_i(\theta) = \mathbb{E}_{\substack{(x,\cdot)\sim p(\cdot) \\ y\sim\pi_\theta(\cdot|x)}}[b_i - r_i(y|x)]$.

### 4.1   TYPES OF CONSTRAINTS

While we write the constraints in Eq. (4) as expectation/average constraints, other forms exist. For instance, uniform constraints impose a minimum reward on every generated (prompt, response) pair:

$$r_i\left(y|x\right) \ \geq \ b_i \text{ for all } (x,\cdot) \sim p\left(\cdot\right), \ y \sim \pi_\theta\left(\cdot|x\right) \text{ and all } i \in \{1, 2, \ldots, k\}. \tag{6}$$

Additionally, chance constraints bound the probability of the inequality holding away from zero:

$$\mathbb{P}_{\substack{(x,\cdot)\sim p(\cdot) \\ y\sim\pi_\theta(\cdot|x)}}\left[r_i\left(y|x\right) \ \geq \ b_i\right] \ \geq \ 1 - \epsilon_i \text{ for all } i \in \{1, 2, \ldots, k\}. \tag{7}$$

These constraints are not equivalent, but they are related. We can rewrite Eq. (7) in the form of average constraints using $1 - \epsilon_i$ as the threshold and taking the expectation of the indicator $\mathbb{1}\{r_i(y|x) \geq b_i\}$. Moreover, Eq. (6) implies Eq. (4), but the converse is not true. Unfortunately, Eq. (6) is difficult to achieve in practice, especially when the data distribution is unknown.

We continue using expectation/average constraints, but similar discussions can extend to other types.

### 4.2   LAGRANGE MULTIPLIERS

We can introduce Lagrange multipliers $\lambda_i \geq 0$ for the constraints and obtain the Lagrangian:

$$\mathcal{L}\left(\theta\right) \ = \ L\left(\theta\right) \ + \ \sum_{i=1}^{k} \lambda_i C_i\left(\theta\right). \tag{8}$$

There is a rich literature connecting the Lagrangian with constrained optimization. Notably, the KKT conditions (Karush, 1939; Kuhn & Tucker, 1951) provide sufficiency conditions for global optimality under convexity, where the solution is obtained by finding the saddle point of the Lagrangian. However, these conditions are not enough for highly non-convex scenarios such as ours.

Nevertheless, the Lagrangian is instructive in understanding the relative importance of the constraints. For an active constraint, i.e., one satisfied with equality, the corresponding Lagrange multiplier can be non-zero; the larger its value, the more important the constraint. Conversely, for an inactive constraint, i.e., one satisfied with strict inequality, the corresponding Lagrange multiplier must vanish to 0. This is known as complementary slackness and is one of the KKT conditions.

### 4.3   LOGARITHMIC BARRIER

A practical way to enforce constraints is with barrier functions. Consider the (relaxed) log barrier:

$$\mathcal{B}_{\mu,s}\left(z\right) \ = \ \begin{cases} -\mu\log\left(-z\right), & z \ \leq \ -s \\ \frac{\mu}{s}z + \mu - \mu\log s, & z \ > \ -s \end{cases}, \text{ and hence } \partial_z\mathcal{B}_{\mu,s}\left(z\right) \ = \ \frac{\mu}{\max\left(-z, s\right)}, \tag{9}$$

with parameters $\mu, s > 0$. This is a convex, continuous, and differentiable function, which is valid for all $z \in \mathbb{R}$. Importantly, for $s = \mu^2$, this barrier function converges to the characteristic function $\chi\{z \leq 0\}$ as $\mu \to 0$, i.e., it takes the value 0 when $z \leq 0$ and $\infty$ otherwise (Tal et al., 1992; Nash et al., 1994; Hauser & Saccon, 2006; Feller & Ebenbauer, 2017); the condition $s = \mu^2$ is sufficient, but not necessary (Kervadec et al., 2022). This convergence to the characteristic function is visually depicted in Fig. 1, showing the change in the log barrier function as we gradually decrease $\mu$.

We can now use the log barrier to enforce the constraints in Eq. (5) and simply add them to the objective. We obtain an *unconstrained* objective, with $\mu$ controlling the strength of the constraints:

$$\mathcal{G}_\mu\left(\theta\right) \ = \ L\left(\theta\right) \ + \ \frac{1}{k}\sum_{i=1}^{k}\mathcal{B}_{\mu,\mu^2}\left(C_i\left(\theta\right)\right). \tag{10}$$

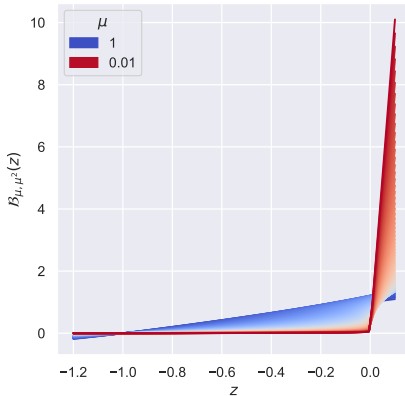

Figure 1: **The relaxed logarithmic barrier.** We depict the convergence of the relaxed logarithmic barrier $\mathcal{B}_{\mu,\mu^2}(z)$ to the characteristic function $\chi\{z \leq 0\}$ as $\mu \to 0$. We gradually decrease $\mu$ from 1 (blue) to 0.01 (red). Consequently, $\mathcal{B}_{\mu,\mu^2}(z)$ gets closer to 0 for $z \leq 0$ and increases to $\infty$ otherwise.

## 5 LAGRANGE LARGE LANGUAGE MODELS (L3MS)

Thus far, we have formulated the SFT and alignment stages as a constrained optimization problem in Eq. (5). We proceed to find solutions for the same by solving the unconstrained objective in Eq. (10). We call the family of models obtained in this way L3Ms, i.e., Lagrange Large Language Models.

### 5.1 OPTIMIZATION PROCEDURE

Since the log barrier converges to the characteristic function as $\mu \to 0$, we want to find the minimizer of $\mathcal{G}_\mu(\theta)$ for a very small $\mu$. However, doing so directly leads to instabilities as the objective function is ill-conditioned. Instead, it is common practice to follow an iterative procedure: one finds the minimizer for a fixed $\mu$, reduces $\mu$, and repeats (Curtis et al., 2024). Specifically, the procedure is instantiated with initial values $\theta_0$, $\mu_0$, and $0 < \gamma < 1$. On the $t$-th iteration, $\mu_t \leftarrow \gamma\mu_{t-1}$ is reduced and $\theta_t \leftarrow \arg_\theta \min \mathcal{G}_{\mu_t}(\theta)$ (with initialization $\theta_{t-1}$). In doing so, the constraints are gradually enforced, nudging the LLM to satisfy them over the optimization procedure while avoiding instabilities. As $\{\mu_t\} \searrow 0$, the weights $\{\theta_t\}$ converge to the minimizer of the constrained problem.

It is impossible to minimize $\mathcal{G}_{\mu_t}(\theta)$ exactly in many practical applications. Instead, at each iteration, one can take a single optimization step toward the solution. Doing so is amenable to stochastic gradient methods and mitigates computational overhead: the optimization proceeds as normal while the value of $\mu$ is reduced over the course of the procedure. One can guarantee the convergence of this procedure to the optimal solution in some settings; for example, Curtis et al. (2024) prove convergence when dealing with box constraints. However, convergence in a scenario like ours is not guaranteed. Nevertheless, we will experimentally demonstrate its use for our constrained problems.

We employ stochastic gradient methods and derive the gradient of our objective function directly:

$$\partial_\theta \mathcal{G}_\mu(\theta) = \partial_\theta L(\theta) + \frac{\mu}{k} \sum_{i=1}^{k} \frac{\partial_\theta C_i(\theta)}{\max(-C_i(\theta), \mu^2)}. \tag{11}$$

This follows immediately from Eqs. (9) and (10). Note that the gradients $\partial_\theta C_i(\theta)$'s are also known as policy gradients in reinforcement learning literature. We discuss our strategy for estimating these gradients in Appendix B and refer readers to Schulman et al. (2016) for a more detailed review.

### 5.2 CONNECTION TO LAGRANGE MULTIPLIERS

The log barrier and the Lagrangian are intrinsically connected; this becomes evident when comparing Eq. (11) with the (gradient of the) Lagrangian in Eq. (8). In particular, we define the multipliers:

$$\hat{\lambda}_i = \frac{\mu}{k \max(-C_i(\theta), \mu^2)},$$

corresponding to the gradients $\partial_\theta C_i(\theta)$'s in Eq. (11). They can be interpreted as Lagrange multipliers: for active constraints, $\hat{\lambda}_i = 1/k\mu$ is non-zero; for inactive constraints, $\hat{\lambda}_i = -\mu/kC_i(\theta)$ vanishes to 0 as $\mu \to 0$. Hence, the KKT complementary slackness condition is satisfied by design.

## 5.3 MEMORY AND TIME COMPLEXITY

L3Ms differ from traditional LLMs only in the fine-tuning process. In fact, L3Ms require less memory and are faster to fine-tune compared to traditional LLMs. This comes from merging the SFT and alignment stages as we did in our constrained optimization formulation in Eq. (5). We minimize the task objective directly and avoid the need to compute deviations away from the SFT model (for instance, to impose the KL divergence penalty). As a result, we save on loading the SFT model into memory and evaluating it on generated responses at each optimization step. Also, the L3M's log barrier parameter $\mu$ is adjusted during fine-tuning itself, without needing any extra steps.

## 5.4 IMPLEMENTATION DETAILS

**Alternating the objective and gradient clipping**   Gradient clipping is a simple yet effective way to ensure stable training of large models (Goodfellow et al., 2016; Zhang et al., 2020). We employ this technique, albeit with the modification of clipping the gradients of both the task objective and the constraints separately, as they can have varying magnitudes. We achieve this by alternating between reducing the task objective and enforcing the constraints by flipping a fair coin to select one or the other. While this doubles the number of steps to achieve the same effect, it does not increase the amount of work done as now only one part of the objective or the other is evaluated at each step.

**Length normalization**   The gradient of our objective function in Eq. (11) involves the LLM's log-likelihoods on the generated responses through the gradients $\partial_\theta C_i(\theta)$'s (cf. Eq. (12)). To avoid a response length bias, we length-normalize the log-likelihoods, akin to the definition of perplexity.

**Estimating the mean preference rewards**   We need to estimate the expectations involved in the gradient of our objective function in Eq. (11). The expectations in the numerators can be estimated with the per-mini-batch Monte Carlo averages (Mohamed et al., 2020). However, $C_i(\theta)$'s in the denominator need careful consideration. Note that: (i) $C_i(\theta)$ does not involve the gradient, so its estimate can include information from previous mini-batches to reduce the estimate's variance, and (ii) since the weight $\theta$ is updated during fine-tuning, $C_i(\theta)$ is non-stationary. Hence, we use an exponential moving average estimate for the mean (offset) preference rewards $C_i(\theta)$ in the denominator.

## 6 EXPERIMENTAL RESULTS

To illustrate the customization of L3Ms, we empirically evaluate them on: (i) satisfaction of the imposed constraints, and (ii) minimization of the task objective. Our code, based on the Transformers library (Wolf et al., 2020), is available at: `https://github.com/Guneet-Dhillon/l3m`.

## 6.1 SETUP

We use LLaMA-7B (Touvron et al., 2023) for all our experiments, as it is a lightweight LLM pretrained on a large corpus. We are interested in the task of instruction-following, for which we use UltraChat (Ding et al., 2023), a large-scale dataset of instructional conversations. We run all experiments on NVIDIA H100s. Further details of our experimental setup are included in Appendix A.

We refer to the LLM fine-tuned to minimize the SFT objective only (without the alignment stage) as the *SFT* model. We fine-tune LLMs using the minimax approach to find a saddle point of the Lagrangian, as proposed by Moskovitz et al. (2024); Dai et al. (2024), and refer to them as *MMs*. Lastly, we refer to the LLMs fine-tuned using our proposed approach as *L3Ms*.

In what follows, we use different preference reward functions and vary the custom constraint requirements. All results are obtained on a held-out test dataset (not seen during training or validation).

| LLM type | Length | Perplexity |
|---|---|---|
| SFT | 121.6 | $0.805_{\pm0.3}$ |
| L3M [ 50 , 100 ] | 81.3 | $0.804_{\pm0.3}$ |
| L3M [100 , 150 ] | 120.7 | $0.804_{\pm0.3}$ |
| L3M [ 50 , 75 ] | 64.4 | $0.807_{\pm0.3}$ |
| L3M [ 75 , 100 ] | 88.2 | $0.808_{\pm0.3}$ |
| L3M [100 , 125 ] | 111.7 | $0.810_{\pm0.3}$ |
| L3M [125 , 150 ] | 126.5 | $0.809_{\pm0.3}$ |
| L3M [ 75 , 87.5] | 82.9 | $0.811_{\pm0.3}$ |
| L3M [ 87.5, 100 ] | 92.7 | $0.809_{\pm0.3}$ |
| L3M [100 , 112.5] | 104.8 | $0.810_{\pm0.3}$ |
| L3M [112.5, 125 ] | 117.3 | $0.810_{\pm0.3}$ |

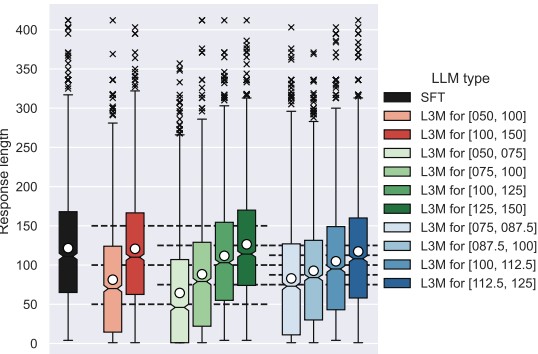

Figure 2: **Length constrained L3Ms.** We report the response lengths (in tokens) and task perplexities of the SFT model and the L3Ms with varying length constraints. *Left:* The mean response length with the mean and standard deviation of the task perplexities. *Right:* The distribution of the response lengths. The notches indicate the medians and their 95% confidence intervals, the boxes show the $\pm25\%$ quantiles, and the whiskers denote the $1.5\times$ interquartile ranges. The white circles mark the means, and the black dashed lines depict the constraints imposed on the different L3Ms.

## 6.2 LENGTH CONSTRAINED L3MS

Consider tasks in which the lengths of the responses need to be contained in the range $[l_{\text{low}}, l_{\text{high}}]$ to control verbosity; for example, in summarization tasks (Makino et al., 2019). In this case, the natural choice for the reward functions compute the response length and its negation: $r_1(y|x) = |y|$ and $r_2(y|x) = -|y|$. Furthermore, these rewards are to be controlled with the minimum requirements of $l_{\text{low}}$ and $-l_{\text{high}}$, respectively. Note that these reward functions are perfectly negatively correlated.

If we naively average the rewards, any unconstrained formulation of alignment (including RLHF) will be ineffective, as the loss will always vanish due to the anti-correlation. We could use a weighted average and tune the weights heuristically, but this is tedious. Instead, we use the constrained formulation and directly constrain the rewards $r_1(y|x) = |y| \geq l_{\text{low}}$ and $r_2(y|x) = -|y| \geq -l_{\text{high}}$.

We fine-tune several L3Ms with varying length constraints. We illustrate the distributions of the generated response lengths (in tokens) and report the perplexities achieved on the task-related data in Fig. 2. We observe that the mean response lengths are in the required range in each case, satisfying the imposed average constraints. Additionally, the task perplexities increase slightly as the constraints on the response lengths are made more stringent. However, there is little to no degradation relative to the SFT model, with all mean task perplexities being within 0.02 standard deviations.

The examples included in Table 1 are generated from such L3Ms. While all the responses correctly answer the prompts, their lengths vary, corresponding to the length constraints imposed on them.

## 6.3 HELPFUL AND HARMLESS L3MS

Next, we consider the Helpful and Harmless (HH) preferences that have been extensively used in the LLM alignment literature (Ji et al., 2023; Wang et al., 2024; Zhou et al., 2024; Guo et al., 2024). Specifically, we utilize the datasets by Bai et al. (2022) to train two preference reward functions, respectively. These learned reward functions are negatively correlated (Bai et al., 2022; Dai et al., 2024). Furthermore, note that the numerical outputs of both these reward functions are interpreted as ratings such that a higher numerical value indicates higher helpfulness/harmlessness and vice versa.

We fine-tune several L3Ms with varying HH constraints. We compare our L3M approach of using log barriers with that of the minimax optimization by Moskovitz et al. (2024); Dai et al. (2024) to find a saddle point of the Lagrangian (MMs). In our experience, learning the Lagrange multipliers with the latter is extremely sensitive to the choice of the learning rate. Moreover, to avoid loading an additional SFT model during fine-tuning, as is done by Moskovitz et al. (2024); Dai et al. (2024), our implementation of MM minimizes the task objective directly (as is done by L3Ms as well).

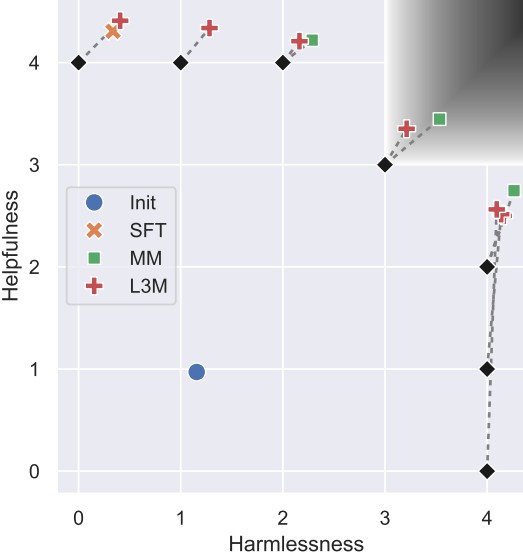

| | Constraints | | Perplexity | |
|---|---|---|---|---|
| **Helpful** | **Harmless** | | **MM** | **L3M** |
| 4 | 0 | | | $0.804_{\pm0.3}$ |
| 4 | 1 | | | $0.809_{\pm0.3}$ |
| 4 | 2 | | $0.822_{\pm0.3}$ | $0.818_{\pm0.3}$ |
| 3 | 3 | | $0.822_{\pm0.3}$ | $0.814_{\pm0.3}$ |
| 2 | 4 | | $0.825_{\pm0.3}$ | $0.820_{\pm0.3}$ |
| 1 | 4 | | | $0.817_{\pm0.3}$ |
| 0 | 4 | | | $0.816_{\pm0.3}$ |

Figure 3: **Helpful and harmless L3Ms.** We report the helpful-harmless rewards and task perplexities achieved by the different LLMs. *Left:* The helpful-harmless rewards attained by the LLM at initialization (at the bottom-left in blue), the SFT model (at the top-left in orange), the MMs (in green), and the L3Ms (in red). We depict the imposed constraints in black, with the dotted gray lines connecting LLMs to their corresponding constraints. Note that constraints are satisfied if the obtained reward point is at the top-right of its corresponding constraint point. For example, the shaded region denotes the feasible region for the constraint point (3, 3), with the shade gradient denoting the distance from the constraint boundary (light to dark shows an increase in distance). *Right:* The mean and standard deviation of the task perplexities for MMs and L3Ms, along with their corresponding constraints; the task perplexity at initialization is $1.316_{\pm0.4}$ and that of the SFT model is $0.805_{\pm0.3}$.

Fig. 3 shows the achieved task perplexities and helpful-harmless rewards for the different LLMs. At initialization, the helpful-harmless rewards are both low, with a high task perplexity of $1.316_{\pm0.4}$. This is improved upon by the SFT model, reducing the perplexity to $0.805_{\pm0.3}$ and attaining a high helpfulness reward (due to the task data instilling instruction-following capabilities). Furthermore, MMs sacrifice task performance to oversatisfy the constraints, with mean task perplexities $\geq 0.822$. Conversely, L3Ms satisfy the imposed helpful-harmless reward constraints with consistently lower task perplexities (the mean perplexities are in the range 0.804-0.820). We attribute this to the L3Ms having better Lagrange multipliers by design rather than learning them as in MMs (cf. Section 5.2).

Note that here we evaluate the LLMs on reward constraint satisfaction and task objective minimization. Furthermore, one can increase the constraint minimum baselines to obtain higher rewards.

## 7 CONCLUSIONS

In this work, we formulate SFT and alignment in LLMs as constrained optimization: we minimize the task objective while simultaneously imposing application-specific constraints on preferences. This enables the customization of LLMs to different preferential properties while maintaining performance on the task of interest. Consequently, we propose Lagrange Large Language Models (L3Ms) to solve this constrained optimization problem by incorporating the constraints in the objective using the logarithmic barrier. We include experimental results to illustrate the customization qualities of L3Ms, which can fit to different preferences, providing a personalized user experience.

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

## A  EXPERIMENTAL SETUP

In addition to the experimental setup discussed in Section 6.1, we provide further details here.

**Task data**  We use UltraChat (Ding et al., 2023), a large-scale dataset of instructional conversations, as our task data to induce instruction-following capabilities. Since each sample contains a sequence of multi-turn question-answer pairs, we randomly sample one of the answers as the response and treat the preceding dialogue as the prompt. We then filter such (prompt, response) pairs to a maximum token length of 512. Consequently, we obtain 340k training samples, 1.7k validation samples, and 1.7 test samples, split randomly since the dataset does not contain train-val-test splits.

**Hyper-parameters**  We fine-tune LLMs for 1 epoch on the task data, with a mini-batch size of 64. We use Adam with a learning rate of $10^{-6}$ and a cosine learning rate scheduler (with 5% of the epoch used for warmup). We set weight decay to 0.1 and the gradient clipping maximum norm to 1. We utilize 16-bit (mixed) precision training and gradient checkpointing. We exponentially decay the log-barrier parameter $\mu$ during fine-tuning from 1 to $10^{-6}$ and use a smoothing factor of 0.1 for the exponential moving average. Lastly, we use top-$p$ sampling ($p$ set to 0.9) for response generation. Apart from this, we use the default hyper-parameters in the Transformers library (Wolf et al., 2020).

### A.1  LEARNING PREFERENCE REWARD MODELS

While some preference reward functions are engineered or rule-based, others are learned. Such preferences can often be difficult to quantify. Alternatively, it is easier to compare responses with respect to the preference, e.g., ranking them from most to least helpful. Consequently, the data for learning preference reward models consist of tuples of the form $(x, y_+, y_-)$, where the prompt $x$ is accompanied by two responses $y_+$ and $y_-$, with a preference for the former response over the latter.

The preference reward model is denoted by $r_\phi(\cdot)$ (parameterized by $\phi$). Assuming the Bradley-Terry model (Bradley & Terry, 1952), the model's predicted probability for preferring $y_+$ over $y_-$ is:

$$p_{r_\phi}(y_+ \succ y_-|x) = \sigma(r_\phi(y_+|x) - r_\phi(y_-|x)),$$

with the standard logistic function $\sigma(\cdot)$. Then, the model minimizes the negative log-likelihood:

$$\min_\phi \mathbb{E}_{(x,y_+,y_-)\sim t(\cdot)}\left[-\log p_{r_\phi}(y_+ \succ y_-|x)\right].$$

Taking inspiration from Rafailov et al. (2023), we initialize the preference reward model $r_\phi(\cdot)$ as a pre-trained LLM and set the reward to be its length-normalized log-likelihood. In this way, we utilize the pre-trained model fully, not just its backbone. As the preference reward model is fine-tuned, its log-likelihoods/rewards are updated to differentiate the preferred responses from the rejected ones.

**Helpful and harmless data**  We use the Helpful and Harmless (Bai et al., 2022) preference data to learn two reward models, respectively. We obtain 161k training samples and 9k test samples after filtering the (prompt, response) pairs to a maximum token length of 2024; 3/4-th are for helpfulness while the remaining 1/4-th are for harmlessness. We further set 5% of the training data for validation.

**Hyper-parameters**  We initialize all reward models with LLaMA-7B (Touvron et al., 2023). We fine-tune for 2 epochs with a mini-batch size of 64. We use Adam with a learning rate of $10^{-6}$ and a cosine learning rate scheduler (with 10% of the epoch used for warmup). We set weight decay to 0.1 and the gradient clipping maximum norm to 1. We utilize 16-bit (mixed) precision training and gradient checkpointing. Apart from this, we use the default hyper-parameters in the Transformers library (Wolf et al., 2020). We validate after every 10% of the epoch and save the best model.

## B  POLICY GRADIENT

We are interested in the policy gradients $\partial_\theta C_i(\theta)$'s. Note that while computing gradients with respect to the parameters of a distribution in an expectation, we can use the log-derivative trick:

$$\partial_\theta \mathbb{E}_{x \sim p_\theta(\cdot)}\left[f\left(x\right)\right] = \int dx f\left(x\right) \partial_\theta p_\theta\left(x\right) = \int dx f\left(x\right) \frac{p_\theta\left(x\right)}{p_\theta\left(x\right)} \partial_\theta p_\theta\left(x\right)$$

$$= \int dx f\left(x\right) p_\theta\left(x\right) \partial_\theta \log p_\theta\left(x\right) = \mathbb{E}_{x \sim p_\theta(\cdot)}\left[f\left(x\right) \partial_\theta \log p_\theta\left(x\right)\right].$$

Applying the above to the policy gradient $\partial_\theta C_i(\theta)$ yields:

$$\partial_\theta C_i\left(\theta\right) = \partial_\theta \mathbb{E}_{\substack{(x,\cdot) \sim p(\cdot) \\ y \sim \pi_\theta(\cdot|x)}}\left[c_i\left(y|x\right)\right] = \mathbb{E}_{\substack{(x,\cdot) \sim p(\cdot) \\ y \sim \pi_\theta(\cdot|x)}}\left[c_i\left(y|x\right) \partial_\theta \log \pi_\theta\left(y|x\right)\right], \tag{12}$$

where $c_i(y|x) = b_i - r_i(y|x)$. This is the simplest form of the policy gradient and can be estimated as Monte Carlo averages. We refer readers to Schulman et al. (2016) for a review of other estimates.

