# OpenReview forum: "L3Ms — Lagrange Large Language Models"
_ICLR.cc/2025/Conference — ICLR 2025 Poster_

### Official Review · Reviewer_tPT4 · 2024-10-28

**Soundness:** 3
**Presentation:** 2
**Contribution:** 2
**Rating:** 5
**Confidence:** 3

**Summary:**

The paper suggests to model SFT and alignment into one constrained optimization problem and proposes a method called L3M, which is to optimize such constrained optimization problems via approximate Lagrangian.

**Strengths:**

- The paper suggests to formulate SFT and alignment into one constrained optimization problem and optimize its approximate Lagrangian.

**Weaknesses:**

- The core objective Eq.(10) $\mathcal{G} _ \mu(\theta)=L(\theta)+\frac{1}{k}\sum _ {i=1}^k\mathcal{B} _ {\mu,\mu^2}(C _ i(\theta))$ is equivalent to the lowerbound of multi-objective RLHF objective with induced reward models and a SFT term. Given a reward model $r _ i(x,y)$, a monotonically increasing function $f _ 1(x)=\mathcal{B} _ {\mu,\mu^2}(x)$, and a monotonically decreasing function $f _ 2(x)=b _ i-x$, we can induce a new reward model $r _ i'(x,y)=-f _ 1(f _ 2(r(x,y)))=-\mathcal{B} _ {\mu,\mu^2}(b _ i-r _ i(x,y))$. As $\mathcal{B} _ {\mu,\mu^2}(x)$ is convex,
$$
\begin{split}
\min &\mathbb{E} _ {x,y}[-r _ i'(x,y)]\\\\
=&\mathbb{E} _ {x,y}[\mathcal{B} _ {\mu,\mu^2}(b _ i-r(x,y))] \\\\
\geq&\mathcal{B} _ {\mu,\mu^2}(\mathbb{E} _ {x,y}[b _ i-r(x,y)])\\\\
=&\mathcal{B} _ {\mu,\mu^2}(C _ i(\theta)),
\end{split}
$$
where the inequality holds by jensen’s inequality. It can be extend into multi-objective scenario with a SFT term $L(\theta)$:
$$
\begin{split}
\min &L(\theta)+\mathbb{E} _ {x,y}\left[\frac{1}{k}\sum _ {i=1}^k-r _ i'(x,y)\right]\\\\
=&L(\theta)+\frac{1}{k}\sum _ {i=1}^k\mathbb{E} _ {x,y}[-r _ i'(x,y)]\\\\
\geq&L(\theta)+\frac{1}{k}\sum _ {i=1}^k\mathcal{B} _ {\mu,\mu^2}(C _ i(\theta))=\mathcal{G} _ \mu(\theta),
\end{split}
$$
which is Eq.(10) of the paper.

- For baselines, as Line 355-358 writed: “L3Ms have the same time complexity as traditional approaches as we combine the SFT and alignment stages into one”. Traditional SFT+alignment, having the same training budget with L3M, should be considered as a baseline.

- For benchmarks, more instruction following tasks should be evaluated to demonstrate the generalization of L3M across diverse distribution.

- For evaluations, 1) ppl (perplexity) can not directly reflect the instruction following ability of an aligned LM. 2) As reward hacking is a considerable problem in RLHF, the score of reward models which are involved into L3M training is not a valid metric. Other metrics should be considered to evaluate the alignment.

**Questions:**

- Why the log barrier parameter $\mu$ should be decayed during training? (Lines 715-716)

- As the log barrier parameter $\mu$ is a hyper-parameter of L3M, what the relationship between its value and L3M performance?

---

> ### Author Response · Authors · 2024-11-21
> **Official Comment by Authors (Part 1 of 2)**
>
> Thank you for your feedback on our work. There seem to be a few misunderstandings, which we clarify below.
>
>
> > [Weakness 1] The core objective [...] is equivalent to the lower bound of [a] multi-objective RLHF objective with induced reward models and [an] SFT term.
>
> You rightly point out that our proposed objective $\mathcal{G}_{\mu} ( \theta )$ (in Eq. 10) is a lower bound to the combined SFT and RLHF alignment objectives with specially chosen reward functions. However, we are unsure why this is considered a weakness of our proposed framework. When minimizing an objective is intractable, a (tight) upper bound is sought after to minimize instead. However, we minimize the desired objective directly in our paper. This analysis further validates that our proposed objective is distinct from simply combining the SFT and RLHF alignment objectives, even with specially chosen reward functions.
>
>
> > [Weakness 2] Traditional SFT+alignment, having the same training budget with L3M, should be considered as a baseline.
>
> Traditional SFT+alignment methods are unfit for constraint satisfaction use cases as they require trial and error to find suitable preference weights; Section 2.4 discusses this in detail. This heuristic process entails training several LLMs on different preference weights. Training a single such LLM has a comparable training budget to that of L3M; however, doing so multiple times is extremely expensive and is not a viable baseline. Instead, we compare our proposed L3Ms with the approach by Moskovitz et al. (2024) and Dai et al. (2024), which has a comparable training cost.
>
>
> > [Weakness 3] For benchmarks, more instruction-following tasks should be evaluated to demonstrate the generalization of L3M across [a] diverse distribution.
>
> We use UltraChat (Ding et al., 2023) for our experiments, which is the state-of-the-art LLM instruction-following dataset. Additionally, we focus on incorporating diverse constraints and demonstrating the versatility of our proposed L3Ms in achieving tailored alignments. We do so by choosing different length constraints and helpful-harmless constraints.
>
>
> > [Weakness 4] For evaluations, 1) ppl (perplexity) can not directly reflect the instruction-following ability of an aligned LM. 2) As reward hacking is a considerable problem in RLHF, the score of reward models [that] are involved [in] L3M training is not a valid metric. Other metrics should be considered to evaluate the alignment.
>
> Our goal is not to develop state-of-the-art LLMs on various performance benchmarks. Instead, we want to build LLMs that satisfy custom reward constraints and simultaneously maintain task performance. We need to decouple the rewards and the task to evaluate this sufficiently.
>
> The only way to test the former is to analyze the rewards obtained by our proposed L3Ms and check if the imposed constraints are satisfied. Reward-hacking is a tangential issue, which can be mitigated through such custom constraints (Moskovitz et al., 2024).
>
> We compute the perplexity on held-out task data to measure task performance. Other metrics that we are aware of involve human or LLM evaluations. For instance, we considered using AlpacaEval (Li et al., 2023; Dubois et al., 2023, 2024). However, such evaluations do not necessarily decouple the task from the rewards.
>
>
> > [Question 1] Why [should] the log barrier parameter $\mu$ [...] be decayed during training?
>
> This is at the very heart of our approach. Apologies for maybe not being clear enough in our detailed discussion in Section 5.1. At optimality, the KKT complementary slackness condition dictates that (constraint x Lagrange multiplier) = 0. Using the log-barrier is a well-established technique in constrained optimization to accomplish this goal. In particular, the log-barrier converges to the characteristic function as $\mu \rightarrow 0$ and we want to find the minimizer of $\mathcal{G}_{\mu} ( \theta )$ for a very small $\mu$. However, doing so directly leads to instabilities as the objective function is ill-conditioned. Instead, we follow an iterative procedure where we take a single optimization step for a fixed $\mu$, reduce $\mu$, and repeat (Curtis et al., 2024). In doing so, $\mu$ decays during training.
>
>
> > [Question 2] As the log barrier parameter $\mu$ is a hyper-parameter of L3M, what [is] the relationship between its value and L3M performance?
>
> We require the log-barrier parameter $\mu$ to decay during training. For the experiments in the paper, we decay $\mu$ from 1 to 1e-6 (Appendix A). We also performed experiments reducing it to 1e-8 instead and observed no change in performance. This is precisely where the benefits of using tools from constrained optimization shine compared to relying on heuristic processes to balance multiple SFT/alignment objectives.

---

> > ### Comment · Reviewer_tPT4 · 2024-11-23
> >
> > Thank you for your reply and explanation. Here are my remaining concerns.
> >
> > > Traditional SFT and alignment methods are unfit for constraint satisfaction use cases as they require trial and error to find suitable preference weights; Section 2.4 discusses this in detail. Such a heuristic process entails training several LLMs on different preference weights. Training a single such LLM has a comparable training budget to that of L3M; however, doing so multiple times is extremely expensive and is not a viable baseline. Instead, we compare our proposed L3Ms with the approach by Moskovitz et al. (2024) and Dai et al. (2024), which has a comparable training cost.
> >
> > The frontier of Traditional SFT+alignment with different weighted rewards, should be considered as a baseline of the frontier of L3LM in the Figure 3 of the paper. This comparison is frontier versus frontier. Besides, in reasonable suspicion, the enforced constraints of L3LM might be broken in some scenarios such as (harmlessness=3, helpfulness=4), (harmlessness=4, helpfulness=3), harmlessness≥5 and helpfulness≥5. If so, in these scenarios, a trial and error process is still needed for L3LM to find suitable constraints to get a satisfied aligned model. The heuristic-driven process is not avoided.
> >
> > > We compute the perplexity on held-out task data to measure task performance. Other metrics that we are aware of involve human or LLM evaluations. For instance, we considered using AlpacaEval (Li et al., 2023; Dubois et al., 2023, 2024). However, such evaluations do not necessarily decouple the task from the rewards.
> >
> > Perplexity can not directly reflect the instruction-following ability of an aligned LLM. Even if the constraints are satisfied, using the trained reward score to measure the performance of LLM might be unreliable because the trained reward model is not the true human reward model (in the paper, no evaluation of the trained reward model is presented in the paper). To evaluate LLM's performance, other metric (such as the win rate between L3LM’s output versus the baseline in the chat dataset) ought to be taken into account.

---

> > > ### Author Response · Authors · 2024-11-26
> > >
> > > Thank you for your continued engagement. We address your queries below.
> > >
> > >
> > > > The frontier of Traditional SFT+alignment with different weighted rewards, should be considered as a baseline of the frontier of L3LM in the Figure 3 of the paper. This comparison is frontier versus frontier. Besides, in reasonable suspicion, the enforced constraints of L3LM might be broken in some scenarios such as (harmlessness=3, helpfulness=4), (harmlessness=4, helpfulness=3), harmlessness≥5 and helpfulness≥5. If so, in these scenarios, a trial and error process is still needed for L3LM to find suitable constraints to get a satisfied aligned model. The heuristic-driven process is not avoided.
> > >
> > > The helpful-harmless constraints experiment (Fig. 3) is __not__ intended to achieve a better rewards frontier but to obtain a better trade-off between constraint satisfaction and the task objective, i.e., satisfying the custom reward constraints (the numerical value being above the requirement) while minimizing the task objective. Our proposed L3Ms achieve this reliably and efficiently. Furthermore, if needed, one could push the L3M reward frontier by increasing the minimum requirements for the helpfulness and harmlessness rewards. We will include this detail in the paper.
> > >
> > > As discussed in Section 2.4, traditional SFT+alignment methods are unfit for constraint satisfaction use cases as they require trial-and-error to find the correct preference weights; this entails training numerous LLMs for a single set of constraints and is extremely expensive. Moskovitz et al. (2024) and Dai et al. (2024) improve on this by proposing to learn the preference weights while training a single LLM. Further, our L3M approach uses preference weights that are optimal by design while also training a single LLM. This implies that if certain constraints are infeasible for L3Ms, they will also be infeasible for the trial-and-error approach. Consequently, we remove the dependence on heuristic processes.
> > >
> > >
> > > > Perplexity can not directly reflect the instruction-following ability of an aligned LLM. Even if the constraints are satisfied, using the trained reward score to measure the performance of LLM might be unreliable because the trained reward model is not the true human reward model (in the paper, no evaluation of the trained reward model is presented in the paper). To evaluate LLM's performance, other metric (such as the win rate between L3LM’s output versus the baseline in the chat dataset) ought to be taken into account.
> > >
> > > We agree that metrics like win rates help evaluate an LLM's overall performance. For this reason, we considered using AlpacaEval (Li et al., 2023; Dubois et al., 2023, 2024). However, such evaluations do not judge based on the task alone but introduce their own bias for preferential properties of the generated responses (Wu & Aji, 2023; Wang et al., 2024).
> > >
> > > These preferential properties are highly application-dependent (as discussed in Sections 1 and 3.1), and one requires LLMs tailored to such custom properties. However, evaluating them using the win rates from before is not useful as both the task performance and the preferential rewards of the LLM influence such a metric. We experience this firsthand as we observe the AlpacaEval win rates of shorter responses to be lower than those for longer ones, even though both the responses are correct (see Table 1 for examples). Instead, we want to decouple the two.
> > >
> > > We compute the perplexity on unseen held-out task data, which measures the (normalized) log-likelihood of the correct responses for the task at hand. This acts as a proxy for the task performance (it is also the default training objective used to perform well on a task). As per evaluating constraint satisfaction, the only way to do so is to analyze the rewards obtained and check if the imposed constraints are satisfied. You are correct that trained reward models do not reflect the true rewards. Because of this, we include two experiments in our paper: (i) with the true reward model for length constraints (the true reward model is known as we can compute the length of the responses without training) and (ii) with the trained reward model for helpful-harmless constraints. Note that training of the reward model is not the novelty presented in our paper.
> > >
> > >
> > > We hope that we have addressed your queries. Please let us know if any questions remain. We would be grateful if you would consider raising your review scores.
> > >
> > >
> > > > Additional References
> > >
> > > Minghao Wu, Alham Fikri Aji. Style Over Substance: Evaluation Biases for Large Language Models. arXiv, 2023.
> > >
> > > Peiyi Wang, Lei Li, Liang Chen, Zefan Cai, Dawei Zhu, Binghuai Lin, Yunbo Cao, Lingpeng Kong, Qi Liu, Tianyu Liu, Zhifang Sui. Large Language Models are not Fair Evaluators. In the Proceedings of the Annual Meeting of the Association for Computational Linguistics, 2024.

---

> > > > ### Comment · Reviewer_tPT4 · 2024-11-27
> > > >
> > > > I would like to thank the authors for providing these additional details. I have adjusted my score accordingly. However, I still maintain that the rigor of the paper needs improvement.

---

> > > > > ### Author Response · Authors · 2024-11-29
> > > > >
> > > > > Thank you again! We incorporated most of the above discussions in our paper revision before the November 27th deadline and will include the remaining ones in the final version. Are there any other queries about the rigor of the paper? We will gladly continue our interaction.

---

> ### Author Response · Authors · 2024-11-21
> **Official Comment by Authors (Part 2 of 2)**
>
> We hope that we have clarified misunderstandings and addressed your queries. We would be grateful if you would consider raising your review scores. Please let us know if any questions remain; we will gladly continue our interaction.
>
>
> > Additional References
>
> Xuechen Li, Tianyi Zhang, Yann Dubois, Rohan Taori, Ishaan Gulrajani, Carlos Guestrin, Percy Liang, Tatsunori B. Hashimoto. AlpacaEval: An Automatic Evaluator of Instruction-following Models. https://github.com/tatsu-lab/alpaca_eval, 2023.
>
> Yann Dubois, Chen Xuechen Li, Rohan Taori, Tianyi Zhang, Ishaan Gulrajani, Jimmy Ba, Carlos Guestrin, Percy S. Liang, Tatsunori B. Hashimoto. AlpacaFarm: A Simulation Framework for Methods that Learn from Human Feedback. In the Proceedings of the Advances in Neural Information Processing Systems, 2023.
>
> Yann Dubois, Percy Liang, Tatsunori Hashimoto. Length-Controlled AlpacaEval: A Simple Debiasing of Automatic Evaluators. In the Proceedings of the Conference on Language Modeling, 2024.

---

### Official Review · Reviewer_oGFz · 2024-11-01

**Soundness:** 4
**Presentation:** 4
**Contribution:** 3
**Rating:** 6
**Confidence:** 3

**Summary:**

This work introduces a new method for fine-tuning and aligning large language models (LLMs) by treating these processes as a constrained optimization problem. Instead of relying on heuristic methods, the authors propose using Lagrange Large Language Models (L3Ms) that incorporate constraints directly into the optimization through logarithmic barriers. This approach allows models to maintain performance on primary tasks while being customized to meet specific user or application-related preferences. The paper provides experimental evidence showing that L3Ms are effective.

**Strengths:**

- Clear problem statement. This work discusses the shortcomings of previous works on constrained RLHF and proposes corresponding improvements to solve it, which seems convincing to me.
- Good optimization problem conversion. I find it quite interesting and instructive how this work converts a general target of balancing task performance and multiple specific requirements into a solvable object function of Lagrangian and introduces the necessary components of the logarithmic barrier to make it practically work.

**Weaknesses:**

- My main concern is with the experimental results in Figure 3, where MM appears to achieve a better performance frontier than L3M. Although L3M results in lower perplexity (PPL), it's unclear how significantly this affects real task performance, as the improvement seems marginal. Could the authors provide further discussion on this?
- The constraints used in the experiments are somewhat limited in terms of variety. It's unclear how the proposed algorithms would perform in scenarios with more complex or demanding constraints.
- The experimental setups seem relatively simple, which may not fully capture the real-world performance of the fine-tuned LLMs.

**Questions:**

- Can authors provided more discussions on computation overheads introduced in this work in comparison to the general SFT and other related works (https://openreview.net/forum?id=TyFrPOKYXw, https://openreview.net/forum?id=gkfUvn0fLU).

---

> ### Author Response · Authors · 2024-11-21
> **Official Comment by Authors (Part 1 of 2)**
>
> Thank you for recognizing our paper's contributions and for your constructive feedback. We address your queries below.
>
>
> > [Weakness 1] My main concern is with the experimental results in Figure 3, where MM appears to achieve a better performance frontier than L3M.
>
> We will discuss this in more detail. Note that when a constraint is satisfied, its influence should vanish (Section 4). Further, only the task objective should be minimized if all constraints are satisfied. Consequently, the helpful-harmless constraints experiment is intended not to achieve a better rewards frontier but to obtain a better trade-off between constraint satisfaction and the task objective.
>
> Fig. 3 shows that the min-max (MM) approach over-satisfies the constraints by sacrificing task perplexity. Conversely, our proposed L3Ms satisfy the constraints while being closer to the constraint line. In doing so, our L3Ms consistently achieve a lower task perplexity. This behavior is attributed to the Lagrange multipliers associated with our approach satisfying the KKT complementary slackness condition by design. Furthermore, one could push the L3M reward frontier by increasing the minimum requirements for the helpfulness and harmlessness rewards.
>
>
> > [Weakness 2] The constraints used in the experiments are somewhat limited in terms of variety. It's unclear how the proposed algorithms would perform in scenarios with more complex or demanding constraints.
>
> In our experiments, we use length constraints and helpful-harmless constraints (with varying constraint requirements in both cases). We also initially considered using the rewards from UltraFeedback (Cui et al., 2024). However, we chose the ones we use because the corresponding rewards are negatively correlated, which is not the case for UltraFeedback. Constraint satisfaction for negatively correlated rewards is harder as it requires finding the right balance between them.
>
> Additionally, we systematically adjust the length constraints to increase difficulty. We do so by decreasing the range of allowed mean lengths from 50 to 25 to 12.5 tokens (roughly equivalent to 38, 19, and 10 words respectively). A permissible range of 10 words is a demanding ask.
>
>
> > [Weakness 3] The experimental setups seem relatively simple, which may not fully capture the real-world performance of the fine-tuned LLMs.
>
> While the experimental setup might seem relatively simple, we use more preference rewards than Moskovitz et al. (2024) and Dai et al. (2024) (with varying constraints), and a 4.5x larger LLM than Moskovitz et al. (2024) – Moskovitz et al. (2024) and Dai et al. (2024) are the most relevant related works.
>
>
> > [Question 1] Can authors [provide] more discussions on computation overheads introduced in this work in comparison to the general SFT and other related works?
>
> Thank you for this question. We will include this discussion in the paper. The short answer is that our proposed L3M fine-tuning is computationally cheaper and more memory-efficient than other methods.
>
> For our analysis, we fix the dataset size to N and the batch size to S. We assume that F is the time taken to compute the logits of a token sequence (the forward pass) and B is the time taken to take an optimization step with respect to the corresponding loss (the backward pass). Additionally, we assume that G is the time it takes to generate a new token sequence.
>
> We first derive the computational cost for typical SFT+alignment methods, which includes the method by Moskovitz et al. (2024) and Dai et al. (2024). Note that these methods carry out the SFT and alignment stages sequentially. SFT minimizes the perplexity on token sequences from task-specific data (Eq. 1), so an epoch of SFT costs N(F+B)/S in time. The alignment stage is more involved. It first generates new token sequences (the responses are generated for task-specific prompts). This is followed by computing the policy gradient corresponding to Eq. 2 and the KL divergence from the reference SFT LLM; this requires two forward passes, one through the LLM being fine-tuned and another through the reference SFT LLM. Finally, the optimization step is taken. As a result, an epoch of alignment costs N(G+2F+B)/S in time. Adding the two costs together, typical SFT+alignment methods (including that by Moskovitz et al., 2024 and Dai et al., 2024) cost N(G+3F+2B)/S per epoch in time.
>
> Conversely, since our L3M approach merges the SFT and alignment stages, we do not need to compute the KL divergence penalty, while the rest remains the same. Therefore, an epoch of L3M costs N(G+2F+2B)/S in time, which is cheaper than other methods. Additionally, our L3Ms avoid loading the reference SFT LLM into memory, making them more memory-efficient than other methods. Note that updating the log-barrier parameter $\mu$ (a real number) at every optimization step is done in constant time. Thank you again for the question, we will include the analysis in the paper.

---

> ### Author Response · Authors · 2024-11-22
> **Official Comment by Authors (Part 2 of 2)**
>
> We hope that we have addressed your queries. We would be grateful if you would consider raising your review scores. Please let us know if any questions remain; we will gladly continue our interaction.
>
>
> > Additional References
>
> Ganqu Cui, Lifan Yuan, Ning Ding, Guanming Yao, Bingxiang He, Wei Zhu, Yuan Ni, Guotong Xie, Ruobing Xie, Yankai Lin, Zhiyuan Liu, Maosong Sun. ULTRAFEEDBACK: Boosting Language Models with Scaled AI Feedback. In the Proceedings of the International Conference on Machine Learning, 2024.

---

> ### Comment · Area_Chair_jZwk · 2024-11-26
> **Reminder: Rebuttal Deadline for ICLR 2025**
>
> Dear Reviewer oGFz,
>
> As the rebuttal deadline approaches, please kindly check the papers' discussion threads and respond to the authors' rebuttals. If you haven't had a chance to respond yet, I’d greatly appreciate your input soon. Your insights are invaluable to the authors and the review process.
>
> Thank you for your effort and support!
>
> Best regards,
>
> Area chair

---

> > ### Author Response · Authors · 2024-11-29
> >
> > Thank you again for your feedback on our work. We hope that we have addressed your queries. We would be grateful if you would consider raising your review scores. Please let us know if any questions remain; we will gladly continue our interaction until the December 2nd discussion deadline.

---

### Official Review · Reviewer_PMWz · 2024-11-01

**Soundness:** 3
**Presentation:** 2
**Contribution:** 3
**Rating:** 5
**Confidence:** 3

**Summary:**

This paper presents a novel approach to the supervised fine-tuning (SFT) and alignment of large language models (LLMs), formulating it as a constrained optimization problem. The authors propose Lagrange large language models (L3Ms), which use logarithmic barriers to enforce application-specific requirements, eliminating the need for heuristic-driven processes. This approach allows for the customization of L3Ms across diverse applications. The paper demonstrates the versatility and efficacy of L3Ms through experimental results, showing their ability to achieve tailored alignments for various applications. This work contributes a new perspective and methodology to the fine-tuning and alignment of LLMs, enhancing their adaptability and effectiveness across different applications.

**Strengths:**

1. The paper innovatively formulates SFT and alignment as a constrained optimization problem, offering a fresh perspective that avoids traditional heuristic methods.
2. L3Ms, with their logarithmic barriers, demonstrate technical excellence and adaptability across diverse applications, ensuring robust performance.
3. By providing a systematic approach to fine-tuning LLMs, this research addresses a critical need and offers practical benefits for both academic and industrial use.

**Weaknesses:**

1. The paper's experimental section may lack comprehensive evaluations, which could limit the full demonstration of L3Ms' capabilities and robustness across a wide range of scenarios.
2. The choice of datasets used in the experiments appears to be restricted, potentially affecting the generalizability of the findings and the ability to assess the method's effectiveness on varied data types.
3. Utilizing LLMs with only 7 billion parameters might restrict the scalability and performance of L3Ms, as larger models are often necessary for handling complex tasks and achieving state-of-the-art results.
4. Writing Issues: The paper has some writing issues, including incorrect formula numbering, which can cause confusion and disrupt the flow of the paper. Improving the clarity and accuracy of the writing would enhance the overall quality and readability of the paper.

**Questions:**

See the above weaknesses.

---

> ### Author Response · Authors · 2024-11-21
>
> Thank you for recognizing our paper's contributions and for your constructive feedback. We address your queries below.
>
>
> > [Weakness 2 + Weakness 1] The choice of datasets used in the experiments appears to be restricted, potentially affecting the generalizability of the findings and the ability to assess the method's effectiveness on varied data types.
>
> We use UltraChat (Ding et al., 2023) for our experiments, which is the state-of-the-art LLM instruction-following dataset. Additionally, we focus on incorporating diverse constraints and demonstrating the versatility of our proposed L3Ms in achieving tailored alignments. We do so by choosing different length constraints and helpful-harmless (Bai et al., 2022) constraints. We also initially considered using the rewards from UltraFeedback (Cui et al., 2024). However, we chose the ones we use because the corresponding rewards are negatively correlated, which is not the case for UltraFeedback. Constraint satisfaction for negatively correlated rewards is harder as it requires finding the right balance between them.
>
>
> > [Weakness 3 + Weakness 1] Utilizing LLMs with only 7 billion parameters might restrict the scalability and performance of L3Ms, as larger models are often necessary for handling complex tasks and achieving state-of-the-art results.
>
> We agree that LLMs larger than 7B achieve state-of-the-art results. That said, we believe that the added value of using a 70B model to demonstrate the relevance of our approach beyond what we showcase with the 7B model will be minor compared to the significant associated cost, requiring multiple 8-way H100 servers for fine-tuning. Note that our model is 4.5x larger than the LLM used by Moskovitz et al. (2024), one of the most relevant related works.
>
>
> > [Weakness 4] Writing Issues.
>
> We have updated the paper to improve clarity and readability.
>
>
> We hope that we have addressed your queries. We would be grateful if you would consider raising your review scores. Please let us know if any questions remain; we will gladly continue our interaction.
>
>
> > Additional References
>
> Ganqu Cui, Lifan Yuan, Ning Ding, Guanming Yao, Bingxiang He, Wei Zhu, Yuan Ni, Guotong Xie, Ruobing Xie, Yankai Lin, Zhiyuan Liu, Maosong Sun. ULTRAFEEDBACK: Boosting Language Models with Scaled AI Feedback. In the Proceedings of the International Conference on Machine Learning, 2024.

---

> ### Comment · Area_Chair_jZwk · 2024-11-26
> **Reminder: Rebuttal Deadline for ICLR 2025**
>
> Dear Reviewer PMWz,
>
> As the rebuttal deadline approaches, please kindly check the papers' discussion threads and respond to the authors' rebuttals. If you haven't had a chance to respond yet, I’d greatly appreciate your input soon. Your insights are invaluable to the authors and the review process.
>
> Thank you for your effort and support!
>
> Best regards,
>
> Area chair

---

> > ### Author Response · Authors · 2024-11-29
> >
> > Thank you again for your feedback on our work. We hope that we have addressed your queries. We would be grateful if you would consider raising your review scores. Please let us know if any questions remain; we will gladly continue our interaction until the December 2nd discussion deadline.

---

> > > ### Comment · Reviewer_PMWz · 2024-11-30
> > > **Response to authors**
> > >
> > > Thank you for your feedback. After careful review and considering the comments of Reviewer tPT4, I decided to maintain my score unchanged.

---

> > > > ### Author Response · Authors · 2024-12-02
> > > >
> > > > Thank you for your continued engagement. If you have any specific pending queries about our work, we will gladly discuss them further.

---

### Official Review · Reviewer_uf9P · 2024-11-04

**Soundness:** 3
**Presentation:** 3
**Contribution:** 3
**Rating:** 6
**Confidence:** 3

**Summary:**

This paper presents Lagrange large language models (L3Ms), which formulate supervised fine-tuning and alignment of large language models as a constrained optimization problem. L3Ms use logarithmic barriers to enforce constraints, allowing for customization across diverse applications without relying on heuristics. The paper demonstrates the versatility and efficacy of L3Ms in achieving tailored alignments.

**Strengths:**

The formulation of SFT and alignment as a constrained optimization problem and the use of logarithmic barriers in L3Ms is a novel approach that has the potential to improve the customization of LLMs for different applications.

The paper provides a solid theoretical foundation for the proposed method, including discussions on constraint types, Lagrange multipliers, and the connection to the KKT conditions.

The experimental results on length-constrained L3Ms and helpful and harmless (HH) L3Ms show that the proposed method can effectively customize LLMs according to different constraints while maintaining performance on the original task.

**Weaknesses:**

The experiments mainly consider length constraints and helpful and harmless preferences. It would be interesting to see how the proposed method performs on a more diverse set of tasks, such as computational and reasoning tasks, and whether it can optimize for task-related metrics specific to those tasks.

The paper primarily focuses on perplexity as an evaluation metric. It would be beneficial to include other metrics to provide a more comprehensive assessment of the performance of L3Ms.Are there any plans to incorporate other evaluation metrics in addition to perplexity to better evaluate the performance of L3Ms? If so, which metrics are being considered and why?
Will the authors explore the application of L3Ms to a wider range of tasks, such as computational and reasoning tasks? How would the method be adapted to optimize for task-related metrics specific to these tasks?

**Questions:**

-Are there any plans to incorporate other evaluation metrics in addition to perplexity to better evaluate the performance of L3Ms?

-Will the authors explore the application of L3Ms to a wider range of tasks, such as computational and reasoning tasks? How would the method be adapted to optimize for task-related metrics specific to these tasks?

---

> ### Author Response · Authors · 2024-11-21
>
> Thank you for recognizing our paper's contributions and for your constructive feedback. We address your queries below.
>
>
> > [Question 1 + Weakness 2] Are there any plans to incorporate other evaluation metrics in addition to perplexity to better evaluate the performance of L3Ms?
>
> We compute the perplexity on held-out task data to measure task performance. Other metrics that we are aware of involve human or LLM evaluations. For instance, we considered using AlpacaEval (Li et al., 2023; Dubois et al., 2023, 2024). However, such evaluations do not necessarily decouple the task from the preferential properties.
>
> We highlight that developing state-of-the-art LLMs on various performance benchmarks is not our goal. Instead, we want to build LLMs that satisfy custom preference constraints and simultaneously maintain task performance. We need to decouple the task and the preferences to evaluate this sufficiently.
>
>
> > [Question 2 + Weakness 1] Will the authors explore the application of L3Ms to a wider range of tasks, such as computational and reasoning tasks? How would the method be adapted to optimize for task-related metrics specific to these tasks?
>
> This is a great suggestion! We focused on incorporating varied preference constraints in our experimental setup but did not consider diversifying the tasks. However, the proposed L3M algorithm can easily extend to new tasks (along with corresponding task-related rewards) without modifications since we do not place any assumptions on the task types.
>
>
> We hope that we have addressed your queries. We would be grateful if you would consider raising your review scores. Please let us know if any questions remain; we will gladly continue our interaction.
>
>
> > Additional References
>
> Xuechen Li, Tianyi Zhang, Yann Dubois, Rohan Taori, Ishaan Gulrajani, Carlos Guestrin, Percy Liang, Tatsunori B. Hashimoto. AlpacaEval: An Automatic Evaluator of Instruction-following Models. https://github.com/tatsu-lab/alpaca_eval, 2023.
>
> Yann Dubois, Chen Xuechen Li, Rohan Taori, Tianyi Zhang, Ishaan Gulrajani, Jimmy Ba, Carlos Guestrin, Percy S. Liang, Tatsunori B. Hashimoto. AlpacaFarm: A Simulation Framework for Methods that Learn from Human Feedback. In the Proceedings of the Advances in Neural Information Processing Systems, 2023.
>
> Yann Dubois, Percy Liang, Tatsunori Hashimoto. Length-Controlled AlpacaEval: A Simple Debiasing of Automatic Evaluators. In the Proceedings of the Conference on Language Modeling, 2024.

---

> ### Comment · Area_Chair_jZwk · 2024-11-26
> **Reminder: Rebuttal Deadline for ICLR 2025**
>
> Dear Reviewer uf9P,
>
> As the rebuttal deadline approaches, please kindly check the papers' discussion threads and respond to the authors' rebuttals. If you haven't had a chance to respond yet, I’d greatly appreciate your input soon. Your insights are invaluable to the authors and the review process.
>
> Thank you for your effort and support!
>
> Best regards,
>
> Area chair

---

> > ### Author Response · Authors · 2024-11-29
> >
> > Thank you again for your feedback on our work. We hope that we have addressed your queries. We would be grateful if you would consider raising your review scores. Please let us know if any questions remain; we will gladly continue our interaction until the December 2nd discussion deadline.

---

> ### Comment · Reviewer_uf9P · 2024-12-02
>
> Thanks for providing the feedback. Since my assessment score is positive, I have decided to still maintain this score.

---

> > ### Author Response · Authors · 2024-12-02
> >
> > Thank you for your continued support of our work!

---

### Author Response · Authors · 2024-11-21

Thank you to all the reviewers for recognizing our paper's contributions and for their constructive feedback. We have responded to each of you individually. Further, we have updated the paper to improve clarity and readability, as suggested by Reviewer PMWz.

---

### Meta-Review · Area_Chair_jZwk · 2024-12-22

**Metareview:**

Summary of the paper: This paper introduces Lagrange Large Language Models (L3Ms), an approach to the SFT and alignment of LLMs by framing these tasks as a constrained optimization problem. Logarithmic barriers are used to enforce application-specific constraints, eliminating the need for heuristic methods and allowing for greater customization across diverse applications. Experimental results demonstrate the versatility and efficacy of L3Ms, showcasing their ability to achieve tailored alignments while maintaining strong performance on primary tasks.

Strengths of the paper:
- Well-Motivated: The paper identifies shortcomings in existing constrained RLHF methods and proposes convincing improvements, making the optimization problem more practical and solvable.
- Innovative Approach and Theoretical Foundation: The formulation of SFT and alignment as a constrained optimization problem, utilizing logarithmic barriers in L3Ms, offers a fresh perspective and a solid theoretical basis, including discussions on constraint types, Lagrange multipliers, and KKT conditions.
- Experimental Validation: The experimental results effectively demonstrate the method's capability to customize LLMs under various constraints while maintaining performance on original tasks and avoiding heuristic-driven processes.

Weaknesses of the paper: All reviewers share similar concerns regarding the evaluation:
- Limited Datasets and Tasks Evaluated (Reviewers uf9P, PMWz, oGFz, tPT4): The choice of datasets/tasks in the experiments appears limited, which may affect the generalizability of the findings and hinder the assessment of the method's effectiveness across different scenarios. L3Ms are recommended to be evaluated on a broader variety of tasks, such as computational and reasoning challenges.
- Need for More Evaluation Metrics (Reviewers uf9P, oGFz, tPT4): Incorporating additional evaluation metrics beyond perplexity could be beneficial, providing a more comprehensive understanding of L3Ms' performance.
- Insufficient Results on Larger Models (Reviewer PMWz): Using models with only 7 billion parameters in the experiment may limit the scalability and performance of L3Ms.

Reasons for the Decision
During the rebuttal, the authors primarily addressed the conceptual issues raised by the reviewers. I believe that most of these conceptual concerns have been adequately resolved after reviewing all the comments. However, the authors did not include additional experimental results as suggested by the reviewers. After considering the authors' responses, I am partially convinced of their reasoning for not using evaluation metrics beyond perplexity (as discussed with Reviewer tPT4) and for not performing experiments on larger models (Reviewer PMWz). Nonetheless, I still believe that the authors should evaluate the proposed method across a wider range of tasks to empirically justify their claim that "the proposed L3M algorithm can easily extend to new tasks (along with corresponding task-related rewards) without modifications since assumptions on the task types are not placed." Upon careful consideration, I am slightly leaning towards accepting this paper. It is well-motivated, and the proposed method is innovative with a solid theoretical foundation. More importantly, this work contributes a new perspective to the alignment of LLMs, which could provide valuable insights for the research community. Nevertheless, I strongly recommend that the authors include more empirical evidence to support their method in the camera-ready version.

**Additional Comments On Reviewer Discussion:**

As discussed above, the authors primarily addressed the conceptual issues raised by the reviewers during the rebuttal. I believe that most of these conceptual concerns have been adequately resolved after reviewing all the comments. However, the authors did not include additional experimental results as suggested by the reviewers.

---

### Decision · Program_Chairs · 2025-01-22

Accept (Poster)